# Chronic Urticaria Biomarkers IL-6, ESR and CRP in Correlation with Disease Severity and Patient Quality of Life—A Pilot Study

**DOI:** 10.3390/biomedicines11082232

**Published:** 2023-08-09

**Authors:** Matea Kuna, Mario Štefanović, Blaženka Ladika Davidović, Nikolina Mandušić, Ines Birkić Belanović, Liborija Lugović-Mihić

**Affiliations:** 1Department of Dermatovenereology, University Hospital Center “Sestre Milosrdnice”, Vinogradska 29, 10000 Zagreb, Croatia; nmandusic@gmail.com (N.M.); ines.birkic.belanovic@gmail.com (I.B.B.); liborija@sfzg.hr (L.L.-M.); 2Department of Clinical Chemistry, University Hospital Center “Sestre Milosrdnice”, Vinogradska 29, 10000 Zagreb, Croatia; mario.stefanovic@kbcsm.hr; 3Department of Oncology and Nuclear Medicine, University Hospital Center “Sestre Milosrdnice”, Vinogradska 29, 10000 Zagreb, Croatia; blazenka.ladika@kbcsm.hr

**Keywords:** chronic urticaria, biomarkers, urticaria activity score, inflammation, disease burden, once-daily UAS, quality of life (QL)

## Abstract

(1) Background: To assess the relationship between serum interleukin-6 (IL-6), erythrocyte sedimentation rate (ESR) and C-reactive protein (CRP) values and disease severity in patients with chronic spontaneous urticaria (CSU) and to examine which of these serum biomarkers better indicates disease severity. (2) Methods: Our pilot study included 20 patients with CSU who filled out questionnaires concerning disease severity and quality of life (the Urticaria Activity Score summed over 7 days [UAS7], the once-daily Urticaria Activity Score [UAS], the Urticaria Control Test [UCT], and the Dermatology Life Quality Index [DLQI]). Blood samples were taken to measure IL-6, ESR and CRP. (3) Results: ESR significantly correlated with the UAS7 (linear and moderate correlation; r = 0.496; *p* = 0.026), while CRP did not correlate with disease severity. IL-6 correlated with the once-daily UAS (r = 0.472; *p* = 0.036) and DLQI (r = 0.504; *p* = 0.023) (linear and moderate correlation) but not the UAS7 or UCT. (4) Conclusions: IL-6 was a better indicator of the once-daily UAS and DLQI, while ESR was a better indicator of the UAS7 (there was no correlation between IL-6, CRP and ESR parameters). Although our results are promising, this study should be conducted with a larger number of CSU patients.

## 1. Introduction

Chronic urticaria (CU) is an inflammatory skin disease characterized by hives, angioedema, or both lasting at least 6 weeks, accompanied by itching [1,2]. By etiology, CU is classified into chronic spontaneous urticaria (CSU) (no known cause), and chronic induced urticaria (CIU), triggered by various physical or non-physical (environmental) stimuli [2]. The overall lifetime prevalence of CU is 4.4%, and the point prevalence of CU (1-year prevalence in most studies) ranges from ≤1.5% in the USA and Europe to 3–4% in Mexico, Korea and China [3]. The incidence of CU in the general population is about 1.4% per year [4]. The average age of adult patients with CSU is 30–70 years, and the age of disease onset is 30–50 years, with a higher prevalence in women [5]. Pediatric CSU is considered to be much less prevalent than in adults with an overall prevalence of 0.1% to 0.3% may be even higher as CSU is the most common form of CU in the pediatric population [6].

In most cases, CU is a self-limiting disease that lasts two to five years, and in 20% of patients, more than 5 years [2,7]. In pathogenesis, the occurrence of hives and angioedema involves mast cell and basophil activation by IgE antibodies against its high-affinity receptor (FcϵRI) or IgG antibodies against IgE/FcϵRI and the release of several mediators (i.e., histamine, tryptase, leukotriene C4 (LTC4), prostaglandin D2 (PGD2), platelet-activating factor (PAF), granulocyte-macrophage colony-stimulating factor (GM-CSF), matrix metalloproteinase-9 (MMP-9), C-X-C motif chemokine ligand 1/2 (CXCL1/2), tumor necrosis factor α (TNFα), IL-1β, IL-6, IL-2, IL-12, IL 17, IL-23, IL-13, IL-18, etc.) [8] which results in the vasodilation and extravasation of plasma and the activation of sensory nerves, leading to hives and itching [9,10,11,12].

In clinical work, it is very important to look at the relevant disease biomarkers. According to the latest guidelines for CSU treatment and diagnostic procedures, blood work should look for differential blood count (DBC), erythrocyte sedimentation rate (ESR) and/or C-reactive protein (CRP), IgG anti-TPO and total IgE values. An extended workup includes the Helicobacter pylori antigen test, circulating basophils, thyroid hormones and antibodies [1,13]. Other commonly mentioned laboratory factors/parameters and indicators of CSU severity are values for serum cytokine IL-6, D-dimers and vitamin D [14]. However, there have been very few studies examining the relationship between serum CSU biomarkers and CSU severity and patient quality of life, with no specific data comparing biomarkers and disease duration. According to available data, serum CRP, ESR and IL-6 values are elevated during the active stage of CSU, while during remission they are significantly lower [15,16,17,18]. It has also been found that serum IL-6 and CRP values significantly correlate with CSU severity [18,19]. According to a handful of studies, a significant association between serum IL-6 levels and urticaria activity was found (measured by the Urticaria Activity Questionnaire, or UAS7), as well as between IL-6 and impaired quality of life as measured by the DLQI [18,20,21]. It is known that IL-6 is mainly secreted by mast cells and T cells and is a B-cell differentiation factor associated with a Th2 response, involved in the etiopathogenesis of inflammation, hematopoiesis, and immune regulation [22].

To determine CU severity/activity, the UAS7, UCT, and DLQI questionnaire-based indices help define the impact of CU on patients, especially helpful in studies and clinical practice [1,23,24]. The UAS7 is an indicator of CU severity of the previous 7 days and is based on the evaluation of hives and itching [1]. In comparison, the UCT determines CU severity and the degree of disease control over a longer period of time—the previous 4 weeks [23]. There is also the once-daily UAS which measures urticaria activity (itch severity and number of hives) of the previous 24 h, shown by Jauregui et al. to be a reliable instrument for assessing disease activity in clinical practice [25]. To assess the quality of life of dermatological patients, the DLQI is often used, which includes 10 questions that refer to the previous 7 days [24].

Based on all of the above, there is a need to find a more precise serum CSU biomarker that would assess CSU severity and serve to monitor the clinical response during treatment [1]. Thus, we conducted this pilot study to examine the relationship between serum IL-6, CRP and ESR values and CSU severity expressed by questionnaires (the UAS7, the once-daily UAS on the day of blood sampling, the UCT, and the DLQI). In addition, we wanted to determine the correlation between IL-6, CRP and ESR, and which of these serum markers is a better indicator of disease severity as measured by the questionnaires, including the Croatian version of the once-daily UAS.

## 2. Materials and Methods

### 2.1. Subjects

Our cross-sectional pilot study was conducted at the Department of Dermatovenereology, University Hospital Center “Sestre Milosrdnice”, Zagreb, Croatia, in the period between November 2021 and June 2022.

Our cross-sectional pilot study included a total of 20 patients with chronic spontaneous urticaria (CSU), 6/20 men and 14/20 women, mean age of 46 ± 18.4 years; range of 32–67 years. The mean disease duration was 13.6 ± 15.2 months (range 5–21 months), who were regularly treated with antihistamines (bilastin was administered orally in the form of tablets at a dose of 20 mg twice daily, and in worsening up to 4 tbl daily).

Inclusion criteria were ≥18 years of age with a diagnosis of CSU, defined as the daily, or almost daily, occurrence of generalized hives or angioedema for at least 6 weeks prior to inclusion. Patients were excluded from the study if they suffered from acute urticaria, urticaria vasculitis or other forms of urticaria not associated with the chronic form of the disease, any form of inducible CU not associated with CSU, angioedema without hives, any systemic disease or other condition that could hinder data collection or interpretation, as well as patients taking immunosuppressives, or psychoactive therapy, or biologics. Patients taking systemic corticosteroids 2 weeks prior to the study were also excluded.

### 2.2. The Ethical Statement

The study was approved by the Ethics Committee of the University Hospital Center “Sestre Milosrdnice”, Zagreb, Croatia in November 2021 (Number of protocol: 251-29-11-21-01-9), and all patients included in the study gave their written informed consent to participate.

### 2.3. Methods

Each participant’s blood was drawn to determine serum levels of IL-6, ESR and CRP.

Each participant also filled out questionnaires on CSU severity/activity related to different time periods—the once-daily UAS (conducted on the day of blood sampling), the UAS7 (7-day recall period prior to blood sampling), the UCT (4-week recall period prior to blood sampling). Finally, all participants took the DLQI (7-day recall period prior to blood sampling).

#### 2.3.1. Serum Parameters

Serum samples were taken to analyze ESR, CRP and IL-6. ESR was measured using the Westergren method, which measures how far (in millimeters) red blood cells fall, due to gravity, toward the bottom of a standardized, upright, elongated tube over one hour. To measure ESR, blood was collected in Vaccuette^®^ test tubes (Greiner Bio-One, Kremsmünster, Austria) with citrate anticoagulants.

CRP concentration was measured using the original manufacturer’s reagents on the Architect c8000 clinical chemistry analyzer (Abbott Diagnostics, Abbott Park, IL, USA). A latex immunoturbidimetric assay is used, where agglutinates are made after a reaction between CRP in the sample and anti-CRP antibody absorbed by latex particles.

Serum IL-6 was assessed by electrochemiluminescence (ECL) using a Roche Cobas E601 immunology analyzer (reagents Elecsys: IL-6 100T v2, IL-6 calset, diluent multi assay, and precicontrol multimarker).

For serum IL-6, normal referral value concentrations range from 0–7 pg/mL, for CRP from 0–5 mg/L, and for ESR from 5–28 mm/3.6 ks.

#### 2.3.2. Questionnaires

To analyze CSU patients’ disease severity and quality of life, the UAS7, once-daily UAS, UCT and DLQI questionnaires were used according to the new guidelines [1].

The Urticaria Activity Score (UAS) is a sum based on the assessment of urticaria and pruritus separately on a scale from 0–3 during a 24-h period. The scoring of hives is as follows: no hives (0 points), mild (<20 hives, 1 point), moderate (20–50 hives, 2 points), severe (>50 hives, 3 points). Scoring of itching is as follows: no itching (0 points), mild (itching present but subjectively does not present a problem, 1 point), moderate (itching presents a problem but does not interfere with daily activities or sleep, 2 points), intense (severe itching, disrupts daily activities or sleep, 3 points). The sum of the once-daily UAS ranges from 0–6 points and was interpreted for the purposes of this study as 0 = no disease, 1 = well-controlled disease, 2 = mild disease, 3–4 = moderate disease; 5–6 = severe disease (Croatian version of the once-daily UAS). The weekly UAS7, on the other hand, is a sum of 7 consecutive days, ranging from 1–42 points, where 1–6 points indicates well-controlled disease, 7–15 points is mild disease, 16–27 points is medium severity, and scores of 28–42 indicate severe disease [1].

The Urticaria Control Test (UCT) consists of 4 questions referring to the previous 4 weeks, and it examines how frequent the physical symptoms (itching, hives, swelling) of urticaria were during those weeks, how much the urticaria affected quality of life, how often medications were not able to control symptoms, as well as how well the urticaria was under control. The possible answers to each question are as follows: very much (4 points), a lot (3 points), occasionally (2 points), a little (1 point), does not apply to me at all (0 points). The possible total sum is 16 points, with a clearly defined cutoff (a score of 12) for patients with “well-controlled” versus those with “poorly controlled” disease (a score below 12), which helps in making treatment decisions [23].

The Dermatology Life Quality Index (DLQI) has a one-week recall period and contains 10 questions in six main items: symptoms and feelings (questions 1 and 2), daily activities (questions 3 and 4), free time (questions 5 and 6), work and school (question 7), personal relationships (questions 8 and 9) and treatment (question 10). All questions include the following possible answers: very much (3 points), a lot (2 points), a little (1 point), not at all (0 points) and not related in my case (0 points). The sum can range from 1–30, the minimum being 0 to 1 (no impact on the patient’s life), 2 to 5 indicating a small impact, 6–10 being moderate impact, 11–20 meaning very large impact, and the maximum of 21 to 30 is interpreted as having an extremely large impact on the patient’s life) [24].

#### 2.3.3. Statistical Analysis

Data distribution normality was checked with the Shapiro-Wilk test. Spearman’s correlation with Cohen’s criteria for interpretation was used for statistical analysis of the correlation of variables: r = 0.25–0.3 = weak correlation, 0.3–0.5 = moderate, 0.5–0.7 = large and >0.7 = very large. Linear regression was used to predict the dependence of one variable on another. The comparison of biomarkers between disease categories was done using the Mann-Whitney test, and the effect size was quantified using the formula r = Z/√N. For the interpretation of the effect size, the limits of Cohen’s criteria were used. Analyses were performed with the commercial software IBM SPSS v.22.0 (IBM Corp., Armonk, NY, USA) with the significance level set at *p* < 0.05.

## 3. Results

### 3.1. Study Population

Our cross-sectional pilot study included a total of 20 patients with CSU, 6/20 men and 14/20 women (mean age 46 ± 18.4 years; range 32–67 years; the mean disease duration was 13.6 ± 15.2 months; range 5–21 months). All patients were older than 18 years of age, had no autoimmune or chronic diseases and did not use immunomodulators, psychiatric therapy or biologics, though they were regularly treated with antihistamines (bilastin was administered orally in the form of tablets at a dose of 20 mg twice daily, and in worsening up to 4 tbl daily during the study). Among our patients, previously, only 3/20 patients were taking systemic corticosteroids before our analysis; in one patient systemic corticosteroids were discontinued four weeks, in the second two and a half weeks, and in the third two weeks prior to entering the study.

### 3.2. Descriptive Statistics

Descriptive statistics of the sample is presented in Table 1.

As can be seen (Table 1), ERS was in the range of 2–28 (median 6); 7/20 subjects had reduced ESR; the rest were in the reference range. CRP was in the range of 0.2–14 (median 1.3), and 2/20 subjects had increased CRP and IL-6; the rest were in the reference range. UAS7 was in the range of 3–30 (median 13.5), and 9/20 subjects had moderate or severe disease (UAS7 cutoff ≥16), while the rest had mild disease or no urticaria. UCT was in the range of 5–13 (median 9), 3/20 subjects had a well-controlled disease (UCT limit ≥12), and the rest had a poorly controlled disease. DLQI was in the range of 0–19 (median 4), 9/20 respondents had a moderate or large effect (DLQI threshold ≥6); the rest had little or no effect. The UAS7 significantly correlated with DLQI and UCT, but the once-daily UAS did not correlate with them (Table 2).

### 3.3. Results for ESR

A significant positive correlation (linear and moderate, r = 0.496; *p* = 0.026) was seen between CSU patients’ ESR and disease severity as measured by the UAS7 (Table 2). Thus, ESR was related to the UAS7 (one-week recall period before sampling), and the correlation was positive, linear and moderate (r = 0.496; *p* = 0.026). Correlations are listed in Table 2. According to the regression equation (UAS7, ESR), an increase in disease severity (UAS7) by 1 scalar point increased ESR by 0.2 mm/3.6 ks (y = 5.8 + 0.2x (R2 = 0.041) (Figure 1).

### 3.4. Results for IL-6

In our CSU patients, IL-6 levels were not connected to their UAS7 and UCT scores but did significantly correlate with the once-daily UAS and DLQI. A moderate and linear positive correlation was found between both IL-6 and the DLQI scores (impaired quality of life) (r = 0.504; *p* = 0.023) and IL-6 and the daily UAS (on the day of blood sampling) (r = 0.472; *p* = 0.036)—as the severity of the disease increased, so did the serum marker.

According to the regression equation, with each increase in DLQI (impaired quality of life) by 1 scalar point, IL-6 increased by 0.3 pg/mL (y = 2.2 + 0.3x; R2 = 0.195) (Figure 2). Serum IL-6 was significantly higher in patients for whom CU had a moderate or large effect on their quality of life (with a moderate effect size; *p* = 0.036; r = 0.469) (Figure 3).

Also, an increase in disease severity (UAS) by 1 scalar point increased IL-6 by 0.8 pg/mL (y = 2.1 + 0.8x (R2 = 0.241) (Figure 4). There were no significant differences in serum biomarker levels between UAS7 categories (dichotomization with cutoff at 16 scalar points).

Our results did not, however, indicate any relationship between CRP and disease severity, or a correlation between IL-6, CRP and ESR parameters.

## 4. Discussion

In our pilot study, conducted in Croatia, comparing CSU indices and biomarkers, the once-daily UAS was confirmed as a valuable indicator of CSU activity in clinical practice (as previously shown for Spanish-speaking patients) and its reliability/validity was similar to those previously reported for other language versions of the once- and twice-daily UAS variants [25]. For example, the once-daily UAS was validated in an observational study from Germany by Młynek et al., and the twice-daily UAS assessment has been accepted by the United States Federal Drug Administration (FDA) as a patient-reported outcome (PRO) supporting label claims for CSU drugs [26].

In our trial, serum IL-6 best correlated with the once-daily UAS and with impaired quality of life (positive, linear and moderate correlation—as the severity of the disease increased, so did the serum biomarker), similar to results by Zhang et al. and Kasperska-Zajac et al., who found elevated serum IL-6 levels associated with disease activity (UAS7) and impaired quality of life (DLQI) [18,21].

Although our pilot study found a correlation between IL-6 and the once-daily UAS, IL-6 was not significantly connected to the UAS7 and UCT, probably due to the small number of participants enrolled and due to frequent changes in urticaria activity. On the other hand, previous/older research by Kasperska-Zajac et al. did find a significant correlation between serum IL-6 concentration and CU activity (assessed by calculating the cumulative UAS seven days before blood sampling) [16]. The difference between the two study results could be due to differences in the samples—the study by Kasperska-Zajac involved 25/58 (43.1%) patients with mild CU, 19/58 (32.8%) patients with moderate CU, and 14/58 (24.1%) patients with severe urticaria symptoms (based on the weekly UAS), and H1 and H2 antihistamine drugs were withdrawn at least 10 days before blood sampling, while we included a total of only 20 patients, 11/20 (55%) with mild CU and 9/20 (45%) with moderate disease, where bilastin was administered orally in the form of tablets at a dose of 20 mg twice daily, and in worsening up to 4 tbl daily. Kasperska-Zajac et al. also recorded significantly higher serum CRP values than in healthy subjects [median 9.2 vs. 0.74 mg/L]; the median for serum CRP was lower in our study (1.25 mg/L) [21]. In our trial, CRP was not related to CSU severity at all, similar to results obtained by Plavsic et al., who did not find a significant correlation between CRP and CSU activity [27]. The limitations of both studies could be the small number of patients enrolled. Another limitation of our study could be that antihistamine therapy was not suspended like in the study conducted by Kasperska-Zajac. Kasperska-Zajac et al. also showed a significant correlation between serum CRP concentration and CSU activity (as assessed by the weekly UAS) [16]. Also, Rajappa et al. have shown that serum CRP and IL-6 significantly correlate with CSU severity [19]. Similarly, De Montjoye et al. have shown a significant positive correlation between CRP serum levels and CSU activity (based on the UAS7) [28]. In our trial, however, ESR correlated with the UAS7 (positive, linear and moderate correlation), while there was no correlation between CRP and disease severity. Other relevant data, obtained by Kolkhir et al., found that elevated CRP levels were significantly more often recorded than ESR levels, and ESR + (elevated ESR) was only detectable in one-third of the CRP + (elevated CRP) in CSU patients (35% vs. 57%) [29]. Our results mostly support Kolkhir’s, but limitations such as our small number of patients should be taken into account. We should also mention, that no significant correlation between the IL-6, CRP and ESR values was seen in our trial, similar to Ucmak et al., who did not find a significant correlation between serum IL-6 and CRP [17]. According to current recommendations, among the major aims/tasks of a diagnostic workup for CSU patients are the identification of CSU consequences; the assessment of predictors of the disease course and the patient’s response to treatment; monitoring disease activity, impact, and control [1]. Concerning diagnostics, the standard tests are a differential blood count and CRP and/or ESR (in all patients), while total IgE and IgG anti-TPO are recommended in specialist care. Additional diagnostic testing may also be performed as indicated [1]. Sometimes expanded diagnostics are helpful for determining the prognosis of the disease [13]. Considering that urticaria activity varies and frequently changes, according to the latest guidelines (EAACI/GA^2^LEN/EuroGuiDerm/APAAACI guideline), overall CSU activity is best measured by advising patients to document 24-h self-evaluation scores using the once-daily UAS7 for seven consecutive days. This should be used in routine clinical practice to determine disease activity and therapy response. Assessing disease control with the Urticaria Control Test (UCT) is also very useful. It includes a four-week recall period and is a simple four-item tool that clearly defines a cutoff score for “well-controlled” vs. “poorly controlled” disease) [1]. The results of our pilot study support the usefulness of these measurement tools (once-daily UAS, UAS7 and DLQI) together with the assessment of serum IL-6 and ESR.

The literature includes studies from various countries, with prominent results concerning CSU biomarker values and indices. One study from the US and UK showed a near-perfect correlation between changes in CSU signs/symptoms (measured by the UAS7TD) and changes in the DLQI and CU-Q2oL over the same time period [20]. According to research on CSU patients from Poland, their disease activity (UAS), impact (CU-Q2oL) and control (UCT) were strongly correlated, the UCT and CU-Q2oL showing the highest correlation [30]. Authors from Turkey recorded a strong negative correlation between UCT scores and disease activity (UAS28) and the CU-Q2oL [31]. Another study from Iran showed a correlation between results from the Persian UCT and the CU-Q2oL total score and the UAS, suggesting convergent validity [32]. A study from Brazil showed that their UCT results strongly correlated with the UAS7 and UAS28 results, indicating high levels of convergent validity [33]. According to a study from Colombia, the application of international guideline recommendations for CSU led to a high rate of disease control, assessed by scoring severity and patients’ perception of quality of life [34]. However, authors from Germany highlighted a significant discrepancy between recommendations for managing CSU and real-world clinical practice [35].

The literature also discusses IL-6 as a useful biomarker. After an earlier study on CU patients showed that serum IL-6 and CRP values significantly correlate with CSU severity, a newer study on CSU patients found an association between elevated serum IL-6 and hs-CRP and CSU activity [18,19]. Also, Ucmak et al. and Grzanka et al. recorded a significantly higher median for serum IL-6 concentration in CSU patients than in healthy controls (46.57 pg/mL vs. 20.34 pg/mL; *p* < 0.001) [17,36]. Also, Ucmak et al. show a statistically significant association between the once-daily UAS and IL-6 concentration [17]. Also, Jauregui et al. showed that the once-daily UAS is a valid instrument for assessing disease activity in clinical practice for CSU patients (recorded in Spanish-speaking patients) and has similar reliability and validity to other language versions of the once- and twice-daily variants of the UAS [25]. Also, a number of other studies on CU patients have confirmed that serum IL-6 levels are significantly associated with disease activity (UAS7) and impaired quality of life (DLQI) [18,20,21]. In addition, the influence of disease therapy has been exhibited by biomarker values. In one prospective study on CSU patients treated with omalizumab, IL-6 values significantly decreased after therapy [37]. IL-6 is the main indicator of inflammation for chronic inflammatory diseases (including autoimmune diseases, cancers and cytokine storms), which supports the data showing its involvement in CSU pathogenesis [22]. Aside from IL-6, IFN-ϒ (serum levels) has also been found to be elevated in CU patients, which suggests that both Th1 and Th2 type immune responses participate/are implicated in CSU pathogenesis [22,38].

Other (potential) biomarkers of CSU investigated in the literature are circulating basophils. The ratio of basophils in leukocytes and the number of basophils in the peripheral blood of patients with CSU are significantly low compared to healthy donors and patients with atopic dermatitis. Yanase et al. showed in their review that not only basophil-associated biomarkers, such as plasma histamine and total IgE levels but also cytomarkers, such as basophil numbers in peripheral blood, the ratio in leukocytes, responsiveness to stimuli, expression of receptors and location, may also be useful in determining the disease activity [39]. Also, Poddighe and Vangelista’s review mentions that an overall analysis of three pivotal trials (conducted by Gericke et al., Metz et al., and Akdogan et al.) aiming to assess the safety and efficacy of omalizumab in patients with CSU confirmed the increase of circulating basophils in patients treated with omalizumab at the conventional dose [40].

Although our study has limitations (a significantly small number of patients included, 55% of them with mild CSU, and 10% treated with systemic corticosteroids two weeks prior to entering the study); it showed some strengths. The novelty of our pilot study is in the assessment of CSU severity (evaluated by questionnaires) by different durations/periods in comparison with serum inflammatory parameters. Since urticaria activity often changes, overall disease activity is best measured by the daily UAS and serum IL-6 (a better indicator of short-term snapshots of disease severity (once-daily UAS)), and disease impact on quality of life, by the DLQI, as supported by our study. Also, in our pilot study, ESR was found to be a better indicator of disease severity, as measured by the UAS7, than CRP (did not correlate with disease severity). Future studies with more patients, that do not include therapy, need to be conducted to gain more insight into how these parameters are linked to QL and CSU severity.

## 5. Conclusions

Considering the significant limitations of our study which should be taken into account (the small number of subjects, predominantly mild CSU, and few patients recently treated with systemic corticosteroids), our results support using the once-daily UAS to determine CSU severity and the assessment of serum IL-6 in the routine diagnostics of patients with CSU.

## Figures and Tables

**Figure 1 biomedicines-11-02232-f001:**
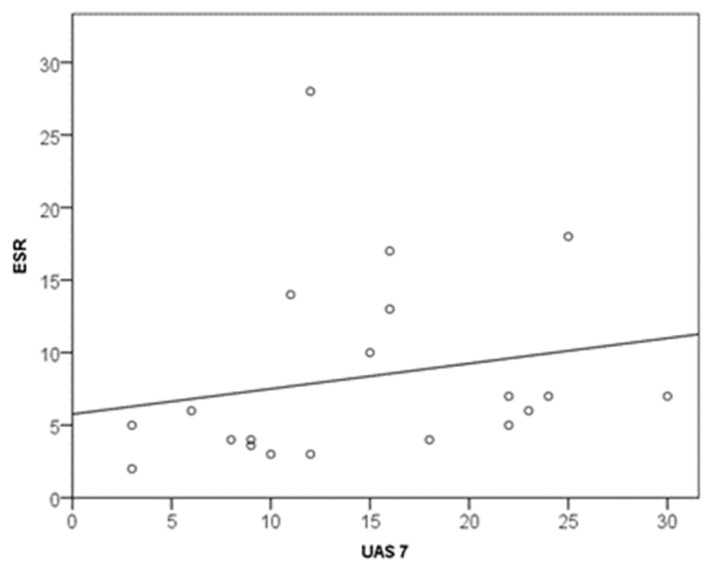
Relationship between UAS7 and ESR.

**Figure 2 biomedicines-11-02232-f002:**
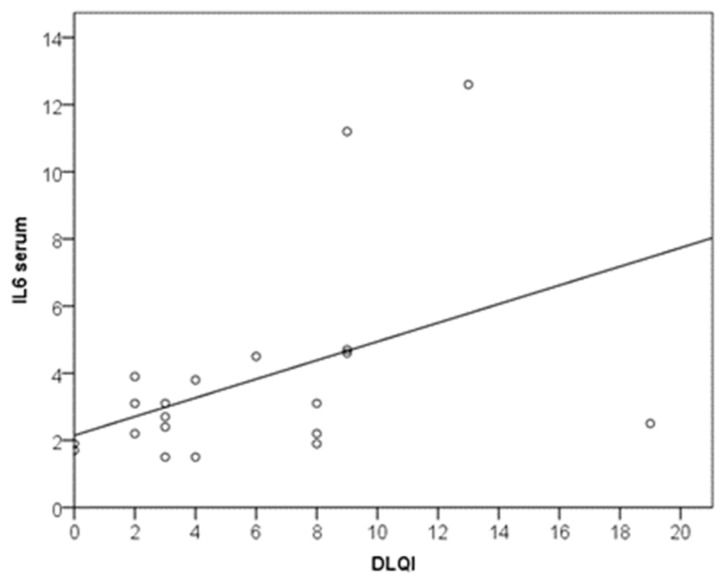
Relationship between DLQI and IL-6.

**Figure 3 biomedicines-11-02232-f003:**
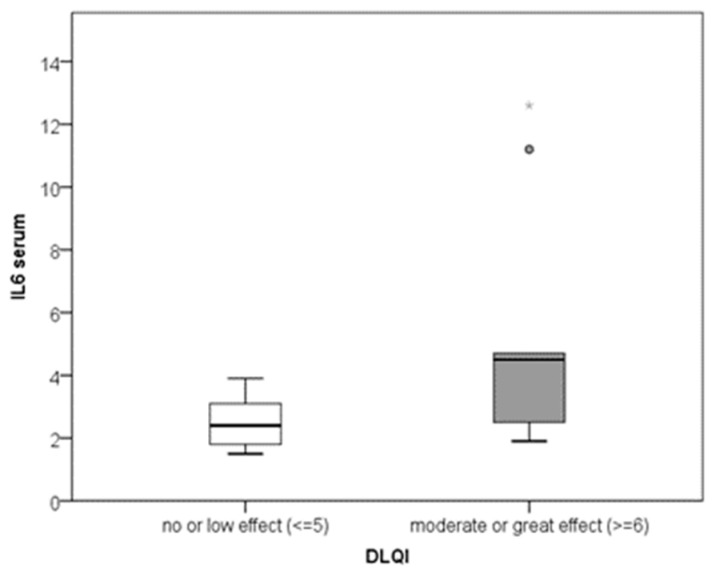
Comparison of serum IL-6 between two categories of quality of life (* the maximum value of serum IL-6).

**Figure 4 biomedicines-11-02232-f004:**
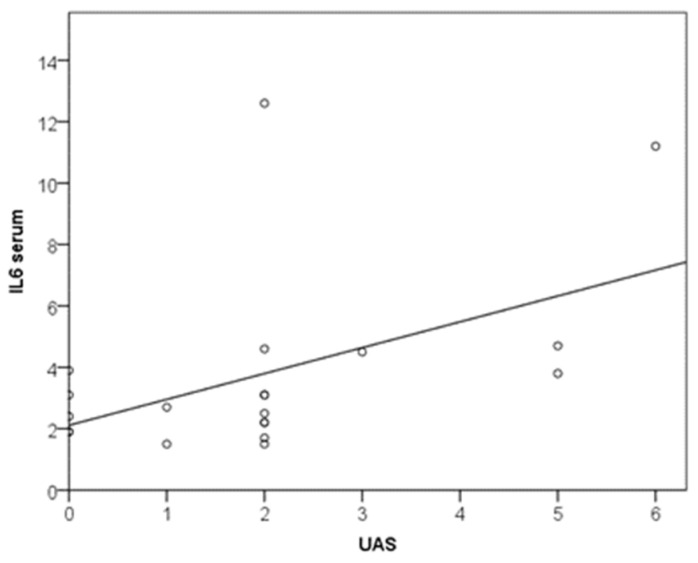
Relationship between UAS and IL-6.

**Table 1 biomedicines-11-02232-t001:** Descriptive statistics.

Variable	Median	Interquartile Range	Minimum	Maximum
IL-6 serum	2.9	2.0–4.4	1.5	12.6
CRP	1.3	0.5–3.0	0.2	14
ESR	6	4–12.3	2	28
Once-daily UAS	2	0.3–2	0	6
UAS 7	13.5	9–22	3	30
DLQI	4	2.3–8.8	0	19
UCT	9	8–10	5	13

**Table 2 biomedicines-11-02232-t002:** Spearman’s correlations between variables (bold values are statistically significant).

		Once-Daily UAS	UAS7	DLQI	UCT	IL-6 Serum	CRP	ESR
Once-daily UAS	R	1	0.271	0.383	0.040	**0.472**	0.022	0.050
	P		0.247	0.096	0.866	**0.036**	0.927	0.833
UAS7	R	0.271	1	**0.784**	**−0.445**	0.332	0.373	**0.496**
	P	0.247		**<0.001**	**0.049**	0.153	0.105	**0.026**
DLQI	R	0.383	**0.784**	1	−0.419	**0.504**	0.403	0.438
	P	0.096	**<0.001**		0.066	**0.023**	0.078	0.054
UCT	R	0.04	**−0.445**	−0.419	1	−0.126	−0.148	−0.107
	P	0.866	**0.049**	0.066		0.597	0.534	0.652
IL-6 serum	R	**0.472**	0.332	**0.504**	−0.126	1	0.214	0.148
	P	**0.036**	0.153	**0.023**	0.597		0.364	0.532
CRP	R	0.022	0.373	0.403	−0.148	0.214	1	0.369
	P	0.927	0.105	0.078	0.534	0.364		0.110
ESR	R	0.050	**0.496**	0.438	−0.107	0.148	0.369	1
	P	0.833	**0.026**	0.054	0.652	0.532	0.110

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
