# Peer review of "Chronic Urticaria Biomarkers IL-6, ESR and CRP in Correlation with Disease Severity and Patient Quality of Life—A Pilot Study"

_biomedicines, 2023, doi:10.3390/biomedicines11082232_

Round 1

Reviewer 1 Report

The authors present an overview of biomarkers in correlation to chronic spontaneous urticaria.

I have some minor comments:

1. Strengths and limitations are missing. The low number is a significant limitation, and this should be pointed out more clearly as well as mention in more detail what conclusions actually can be made.

2. Abbreviations, when first mentioned, should be written in full, also in the abstract, even if very common (ESR, CRP, IL)

Author Response

Response to Reviewer 1 Comments

Point 1: Strengths and limitations are missing. The low number is a significant limitation, and this should be pointed out more clearly as well as mention in more detail what conclusions actually can be made.

Response 1:

Strengths:

     Novelty of our pilot study is using once-daily UAS in compare to other questionnaires concerning disease severity and quality of life (UAS7, UCT, and DLQI).

      In our pilot study ESR is a better indicator of disease severity measured by the UAS7, than CRP (did not correlate with disease severity).

      We have also found that IL-6 is a better indicator of impaired quality of life (DLQI) in patients with CSUi (in our study 8/20 patients had moderate or severe quality of life and 2/20 had eleveted serum IL-6). We have found that IL-6 is a better indicator of short-term snapshots of disease severity (once-daily UAS)

Limitations:

    Significant limitation due to small number of included patients (a total of 20 patients with CSU who were regularly treated with antihistamines (two tablets of bilastin daily).

    Our pilot study did not record/reveal a correlation between IL-6, CRP and ESR probably due to a small number of included patients who were treated with antihitamines during study.

    It is necessary to conduct a study with more subjects who have CSU and who do not use any therapy including antihistamines.

Conclusions: Regardless of the small number of patients, our study has showed some strengths:

-In our pilot study we have found that ESR significantly correlated with the UAS7 (r=0.496; p=0.026), while CRP did not correlate with disease severity at all. This could be explained by ESR largely dependent on the elevation of fibrinogen (an acute phase reactant with a half-life of approximately one week) and also D-dimer (decomposition of fibrinogen) which cause ESR to remain higher for the longer period of time despite the removal of inflammatory stimuli. In contrast, CRP (with a half-life of 6-8 hours) rises rapidly and can quickly return to normal limits (Jain S, 2011) if treatment is used as in our study. Asero et al. have showed that only 50% of patients with severe CSU show elevated D-dimer plasma levels. The activation of the coagulation/fibrinolysis system is associated with a systemic inflammatory milieu [Asero, 2017]. Kasperska-Zajac et al. have showed that CRP concentrations were significantly increased in more severe CU patients when compared to healthy controls and mild CU [Kasperska-Zajac A, 2013], and in our study there were 8/20 patients with moderete, 1/20 with severe and 11/20 patients with mild CSU or well control disease (according to UAS7).

-We have also found that interleukin 6 (IL-6) is a better indicator of impaired quality of life (the DLQI) in patients with CSU (with a moderate effect size; p=0.036; r=-0.469). A number of other studies on CSU patients have also confirmed that serum IL-6 levels are significantly associated with disease activity (UAS7) and impaired quality of life (DLQI) [Zhang, Y 2021; Stull, D.E., 2016; Kasperska-Zajac, 2013]. In our study 8/20 patients had moderate or severe quality of life.

-We have found that IL-6 correlated with the once-daily UAS (r=0.472; p=0.036). This could be due to IL-6 know as an acute phase reactant. In our study 2/20 patients have eleveted serum IL-6 and 18/20 have normal range of serum IL-6.

-Although our pilot study on CSU patients did not record/reveal a correlation between IL-6, CRP and ESR, these results support using the once-daily UAS to determine CSU severity and the assessment of serum IL-6 in the routine diagnostics of patients with CSU.

References:

 (new references have been highlighted)

  1. Jain S, Gautam V, Naseem S. Acute-phase proteins: As diagnostic tool. J Pharm Bioallied Sci. 2011 Jan;3(1):118-27. doi: 10.4103/0975-7406.76489. PMID: 21430962; PMCID: PMC3053509.
  2. Asero R. Severe CSU and activation of the coagulation/fibrinolysis system: clinical aspects. Eur Ann Allergy Clin Immunol. 2019 Oct;52(1):15-17. doi: 10.23822/EurAnnACI.1764-1489.109. Epub 2019 Oct 8. PMID: 31594292.
  1. Kasperska-Zajac, A.; Grzanka, A.; Machura, E.; Mazur, B.; Misiolek, M.; Czecior, E.; Kasperski, J.; Jochem, J. Analysis of procalcitonin and CRP concentrations in serum of patients with chronic spontaneous urticaria. Inflamm Res 2013, 62, 309-312. doi: 10.1007/s00011-012-0580-1.
  2. Zhang, Y.; Zhang, H.; Du, S.; Yan, S.; Zeng, J. Advanced Biomarkers: Therapeutic and Diagnostic Targets in Urticaria. Int Arch Allergy Immunol 2021, 182, 917-931. DOI: 1159/000515753
  3. Stull, D.E.; McBride, D.; Houghton, K.; Finlay, A.Y.; Gnanasakthy, A.; Balp, M.M. Assessing changes in chronic spontaneous/idiopathic urticaria: comparisons of patient -reported outcomes using latent growth modeling. Adv Ther 2016, 33, 214-224. DOI: 1007/s12325-016-0282-0

Point 2: Abbreviations, when first mentioned, should be written in full, also in the abstract, even if very common (ESR, CRP, IL).

Response 2: All abbreviations that appear for the first time in the abstract and in the main text have been written in full and highlighted (erythrocyte sedimentation rate (ESR), C-reactive protein (CRP), interleukin (IL) and ect.

Reviewer 2 Report

INTRODUCTION

- The introduction seems to be long and quite dispersive. Moreover, some important and clear points.

- As regards the prevalence and incidence, also according to age, original and reliable studies should be used to support more precisely this concept.

- It is not clear the use of bold style, which is not consistent with the journal style anyway.

- The introduction should basically provide only the essential background for a better understanding of the study.

- What do you mean exactly as “basophil test”? In general, I would suggest to carefully revise the terminology, wherever this is needed.

METHODOLOGY,

- First of all, the authors should clearly define the study design.

- I would suggest to define better the technical specifications of the laboratory tests.

- A specific subsection for the ethical statement would be useful.

RESULTS

- A first subsection defining the demographic and clinical characteristics of the study population is completely missing.

- Also, the therapeutic aspects should be included.

- The results description should be expanded, overall.

DISCUSSION

- The discussion should start by highlighting the main findings.

- This section seems to be very dispersive. Please, highlight the main findings and orderly discuss them one by one.

- Overall, I think then that the discussion should be extensively reorganized and revised.

- The authors should also discuss other (potential) markers investigated in the literature, in order to highlight the novelty of the current paper. For instance, much attention has been paid to circulating basophils for the disease activity (see: Basophil Characteristics as a Marker of the Pathogenesis of Chronic Spontaneous Urticaria in Relation to the Coagulation and Complement Systems. Int J Mol Sci. 2023 Jun 19;24(12):10320. doi: 10.3390/ijms241210320) and also for the response to biological therapies like omalizumab (refer to: Effects of omalizumab on basophils: Potential biomarkers in asthma and chronic spontaneous urticaria. Cell Immunol. 2020 Dec;358:104215. doi: 10.1016/j.cellimm.2020.104215).

- What about the study limitations??

Moderate editing needed

Author Response

Response to Reviewer 2 Comments

Point 1: INTRODUCTION

- The introduction seems to be long and quite dispersive. Moreover, some important and clear points.

- As regards the prevalence and incidence, also according to age, original and reliable studies should be used to support more precisely this concept.

- It is not clear the use of bold style, which is not consistent with the journal style anyway.

- The introduction should basically provide only the essential background for a better understanding of the study.

- What do you mean exactly as “basophil test”? In general, I would suggest to carefully revise the terminology, wherever this is needed.

Response 1:

-In the revised version of the manuscript, the introduction is shortened with an emphasis on the most important facts about chronic spontaneous urticaria

- The overall lifetime prevalence of CU is 4.4%, and the point prevalence of CU (1- year prevalence in most studies) ranges from ≤1.5% in the USA and Europe to 3–4% in Mexico, Korea and China. (Fricke, J. et al. Prevalence of chronic urticaria in children and adults across the globe: systematic review with meta- analysis. Allergy 75, 423–432 (2020)).

   Average age of adult patients with CSU is ~30–70 years, and age of disease onset ~30–50 years, with more prevalence in women (Kolkhir P, Giménez-Arnau AM, Kulthanan K, Peter J, Metz M, Maurer M. Urticaria. Nat Rev Dis Primers. 2022 Sep 15;8(1):61. doi: 10.1038/s41572-022-00389-z. PMID: 36109590).

- The bold version of the font remained after the last check of the manuscript, in the revised version of the manuscript the font was adjusted according to the journal style

- In the revised manuscript, the introduction is more concise.

- The term "basophil test" is quoted, but in the revised version of the manuscript replaced with term circulating basophils. Terminology has been carefully revised wherever was needed.

Point 2: METHODOLOGY

- First of all, the authors should clearly define the study design.

- I would suggest to define better the technical specifications of the laboratory tests.

- A specific subsection for the ethical statement would be useful.

Response 2:

- the study design: our cross-sectional pilot study included a total of 20 patients with chronic spontaneous urticaria (CSU), 6/20 men and 14/20 women, who were regularly treated with antihistamines (limited to two tablets of bilastin daily)

- Serum samples were taken to analyze IL-6, CRP and ESR.

      ESR was measured using the Westergren method which measures the distance (in millimeters) at which red blood cells fall to the bottom of a standardized, upright, elongated tube over one hour due to the influence of gravity. For measuring estimated sedimentation rate blood was collected in Vaccuette® test tubes (Greiner Bio-One, Kremsmünster, Austria) with citrate anticoagulans.

      Measurement of CRP concentration was performed using original manufacturer’s reagents on Architect c8000 clinical chemistry analyser (Abbott Diagnostics, Abbott Park, IL). The method used is latex immunoturbidimetric assay where agglutinates are made after reaction between CRP in the sample and anti-CRP antibody adsorbed to latex particle.

        Serum IL-6 was assesed by electrochemiluminescence (ECL) using a Roche Cobas E601 immunology analyzer (reagents Elecsys: IL-6 100T v2, IL-6 calset, diluent multy assay, and precicontrol multimarker).    

-the ethical statement: the study was approved by the Ethics Committee of the same hospital in November 2021 (Number of protocol: 251-29-11-21-01-9), and all patients included in the study gave their written informed consent to participate. The revised version of the manuscript contains a special section for the approval of the ethics committee

Point 3: RESULTS

- A first subsection defining the demographic and clinical characteristics of the study population is completely missing.

- Also, the therapeutic aspects should be included.

- The results description should be expanded, overall.

 Response 3:

- our cross-sectional pilot study included a total of 20 patients with chronic spontaneous urticaria (CSU), 6/20 men and 14/20 women, All included patients were older than 18 years, have no autoimmune or chronic diseases and did not use systemic corticosteroids, immunomodulators or psychiatric therapy 2 weeks prior entering the study

- who were regularly treated with antihistamines (limited to two tablets of bilastin daily)

- Descriptive statistics

Table 1. Descriptive statistics

Variable

Median

Interquartile range

Minimum

Maximum

IL6 serum

2.9

2.0-4.4

1.5

12.6

CRP

1.3

0.5-3.0

0.2

14

ESR

6

4-12.3

2

28

UAS daily

2

0.3-2

0

6

UAS 7

13.5

9-22

3

30

DLQI

4

2.3-8.8

0

19

UCT

9

8-10

5

13

7/20 subjects had reduced SE, and the rest were in the reference range. Increased IL6; 2/20 respondents had it, and the rest were in the reference range. The same was true for CRP. Moderate 9/20 respondents had or severe disease (UAS7 cutoff ≥16), and the rest had mild disease or were without urticaria. 3/20 subjects had a well-controlled disease (UCT limit ≥12), a the rest poorly controlled/uncontrolled. Moderate or large effect (DLQI cutoff ≥6); 9/20 respondents had, and the rest had little or no effect. UAS7 significantly correlated with DLQI and UCT, but UAS daily did not correlate with them (Table 2 as presented in the manuscript)

Point 4: DISCUSSION

- The discussion should start by highlighting the main findings.

- This section seems to be very dispersive. Please, highlight the main findings and orderly discuss them one by one.

- Overall, I think then that the discussion should be extensively reorganized and revised.

- The authors should also discuss other (potential) markers investigated in the literature, in order to highlight the novelty of the current paper. For instance, much attention has been paid to circulating basophils for the disease activity (see: Basophil Characteristics as a Marker of the Pathogenesis of Chronic Spontaneous Urticaria in Relation to the Coagulation and Complement Systems. Int J Mol Sci. 2023 Jun 19;24(12):10320. doi: 10.3390/ijms241210320) and also for the response to biological therapies like omalizumab (refer to: Effects of omalizumab on basophils: Potential biomarkers in asthma and chronic spontaneous urticaria. Cell Immunol. 2020 Dec;358:104215. doi: 10.1016/j.cellimm.2020.104215).

- What about the study limitations??

Response 4:

--in the revised manuscript, the discussion begins by highlighting the main findings which are discussed individually

-the discussion has now been reorganized and revised

- approaching to the patients with CSU, it is very important to monitor appropriate biomarkers, so in addition to routine biomarkers (DKS, SE (and/or CRP), anti-TPO IgG and total IgE), extended diagnostics (thyroid hormones and antibodies, circulating basophils), vitamin D, IL-6 and D-dimers are also used as diagnostics tool. In order to emphasize the novelty of the study, we cited the mentioned studies: Basophil Characteristics as a Marker of the Pathogenesis of Chronic Spontaneous Urticaria in Relation to the Coagulation and Complement Systems. Int J Mol Sci. 2023 Jun 19;24(12):10320. doi: 10.3390/ijms241210320 and Effects of omalizumab on basophils: Potential biomarkers in asthma and chronic spontaneous urticaria. Cell Immunol. 2020 Dec;358:104215. doi: 10.1016/j.cellimm.2020.104215

-Limitations:

    Significant limitation due to small number of included patients (a total of 20 patients with CSU who were regularly treated with antihistamines (two tablets of bilastin daily).

    Our pilot study did not record/reveal a correlation between IL-6, CRP and ESR probably due to a small number of included patients who were treated with antihitamines during study.

    Future studies with more patients ,and that do not include therapy, need to be conducted to gain more insight into how these parameters are linked to QL and CSU severity.

Round 2

Reviewer 1 Report

The authors have adequately addressed all reviewers' comments and suggestions

Author Response

Response to Reviewer 1 Comments

Point 1: The authors have adequately addressed all reviewers' comments and suggestions

Response 1: Thank you for your support.

Kind regards,

The Authors

Reviewer 2 Report

INTRODUCTION

- The authors should clearly define CU and CSU, which appears both in the introduction (Chronic spontaneous urticaria guidelines: What is new? J Allergy Clin Immunol. 2022 Dec;150(6):1249-1255. doi: 10.1016/j.jaci.2022.10.004). I mean, define at the first appearance, before describing epidemiology, which could be improved.

- As regards CSU, it can be diagnosed in the pediatric age (0.1-0.3% prevalence, which could be higher, perhaps, as highlighted in: The prevalence of chronic spontaneous urticaria (CSU) in the pediatric population. J Am Acad Dermatol. 2019 Nov;81(5):e149. doi: 10.1016/j.jaad.2019.07.068)

METHODS

- accepted

RESULTS

- Precise therapeutic information in aggregated form should be provided, including dosage regiments. All drugs should be reported, including biologics.

DISCUSSION

- accepted

- some comments following the therapeutic information may be needed.

- strengths and limitation should be part of the discussion

CONCLUSION

- It’s too long. Clear and brief take home messages should be provided.

See above

Author Response

Response to Reviewer 2 Comments

Point 1: INTRODUCTION

- The authors should clearly define CU and CSU, which appears both in the introduction (Chronic spontaneous urticaria guidelines: What is new? J Allergy Clin Immunol. 2022 Dec;150(6):1249-1255. doi: 10.1016/j.jaci.2022.10.004). I mean, define at the first appearance, before describing epidemiology, which could be improved.

- As regards CSU, it can be diagnosed in the pediatric age (0.1-0.3% prevalence, which could be higher, perhaps, as highlighted in: The prevalence of chronic spontaneous urticaria (CSU) in the pediatric population. J Am Acad Dermatol. 2019 Nov;81(5):e149. doi: 10.1016/j.jaad.2019.07.068)

Response 1: In the revsed version of manuscript, CU and CSU are clearly define at the beginning of the Introduction (by Zuberbier 2022), before describing epidemiology.

Also, in epidemiology part, we included the prevalence of chronic spontaneous urticaria (CSU) in the pediatric population.(by Poddighe 2019).

Point 2: METHODS - accepted

Response 2: /

Point 3: RESULTS

- Precise therapeutic information in aggregated form should be provided, including dosage regiments. All drugs should be reported, including biologics.

Response 3: In the revised version of manuscript, we provided precise therapeutic information pertaining to our patients (bilastin was administered orally in the form of tablets at a dose of 20 mg twice daily, and in worsening up to 4 tbl daily). Patients were excluded from the study if they suffered from acute urticaria, urticaria vasculitis or other forms of urticaria not associated with the chronic form of the disease, any form of inducible CU not associated with CSU, angioedema without hives, any systemic disease or other condition that could hinder data collection or interpretation, as well as patients taking immunosuppressives, psychoactive therapy or  biologics. Patients taking systemic corticosteroids 2 weeks prior to the study were also excluded from the study.

Point 4:  

DISCUSSION

- accepted

- some comments following the therapeutic information may be needed.

- strengths and limitation should be part of the discussion

Response 4: We provided all therapeutic information where needed. In the revised manuscript, strengths and limitation are part of the discussion.

Point 5:  CONCLUSION

- It’s too long. Clear and brief take home messages should be provided.

Response 5: We shortened the conclusion, which now provide only clear and brief take home messages.

Our manuscript has been checked by a colleague fluent in English writing, native speaker.

Round 3

Reviewer 2 Report

Overall, the authors addressed most of my comments. I would suggest the authors further improve the results presentation.

Descriptive statistics must be presented in the text, it cannot be linked to the table only. More accurate information on age (mean and standard deviation, range) and clinical aspects (e.g. disease duration, extra-cutaneous manifestations, others) could be included. Results for PCR, ESR and IL-6, do not sound very good in terms of headings. 

The limitations are more than the small sample size. For instance, the use of corticosteroids until 2 weeks before the enrollment is something to be considered. By the way, the authors should provide the numbers of these patients in the results and, in general, the number of patients with a specific therapy.

Finally, the conclusion should be lessened considering the important study limitations. I think the "suggestion" is too strong. 

See above

Author Response

Response to Reviewer 2 Comments

Point 1: Overall, the authors addressed most of my comments. I would suggest the authors further improve the results presentation.

Response 1: Thank you for your support and all suggestions.

Point 2: Descriptive statistics must be presented in the text, it cannot be linked to the table only. More accurate information on age (mean and standard deviation, range) and clinical aspects (e.g. disease duration, extra-cutaneous manifestations, others) could be included. Results for PCR, ESR and IL-6, do not sound very good in terms of headings. 

Response 2: We presented descriptive statistics in the text, as well as in the table. We also included more accurate information on age (mean and standard deviation, range) and clinical aspects (and  disease duration). We rearranged heading of results as well as numbers of tables included.

Point 3. The limitations are more than the small sample size. For instance, the use of corticosteroids until 2 weeks before the enrollment is something to be considered. By the way, the authors should provide the numbers of these patients in the results and, in general, the number of patients with a specific therapy.

Response 3: We expanded the limitation of our study (small numbet of subject, 55% of them with mild CSU, and 10% treated with systemic corticosteroids 2 weeks prior to the study).  

We provided number of patients using systemic corticosteroids (3/20; in one patient systemic corticosteroids were discontinued 4 weeks, in the second 2.5 weeks, and in the third 2 weeks prior to entering the study.

Specific therapy was not included in our study, nor did iur patients usually take any therapy exept antihitamines and 3/20 systemic corticosteroids 2 weeks prior to entering the study.

Point 4. Finally, the conclusion should be lessened considering the important study limitations. I think the "suggestion" is too strong. 

Response 4. We lessened the conclusion considering the important study limitations.

Our manuscript has been checked by a colleague fluent in English writing, native speaker.